# Creatinine Reduction Ratio Is a Prognostic Factor for Acute Kidney Injury following Cardiac Surgery with Cardiopulmonary Bypass: A Single-Center Retrospective Cohort Study

**DOI:** 10.3390/jcm13010009

**Published:** 2023-12-19

**Authors:** Akiko Anzai, Shunsuke Takaki, Nobuyuki Yokoyama, Shizuka Kashiwagi, Masashi Yokose, Takahisa Goto

**Affiliations:** Department of Anesthesiology and Critical Care Medicine, Yokohama City University Hospital, 3-9 Fukuura Kanazawaku, Yokohama 236-0004, Japan; anjy0727@yahoo.co.jp (A.A.);

**Keywords:** reduction of creatinine, acute kidney injury, cardiopulmonary bypass

## Abstract

Acute kidney injury (AKI) after cardiac surgery is a common complication that can lead to death. We previously reported that the creatinine reduction ratio (CRR) serves as a useful prognostic factor for AKI. The primary objective of this study was to determine the predictors of AKI after surgery. The secondary objective was to determine the reliability of the CRR for short- and long-term outcomes. We retrospectively collected information about cardiac surgery patients who underwent cardiopulmonary bypass. Patients were divided into AKI and non-AKI groups based on the AKIN and RIFLE criteria. We analyzed the two groups regarding the preoperative patient data and operative information. The CRR was calculated as follows: (preoperative creatinine—postoperative creatinine)/preoperative creatinine. The prognostic factors of AKI-CS were surgery time, CPB time, aorta clamp time, platelet transfusion, and CRR < 20%. In the multivariate logistical analysis, CRR was an independent predictor of AKI (adjusted odds ratio: 0.90 [0.87–0.93], *p* < 0.001). However, there were no significant differences in CRR in terms of the rate of new onset chronic kidney disease (CKD). After cardiac surgery with cardiopulmonary bypass, CRR has good diagnostic power for predicting perioperative AKI. However, we cannot use it as a prognostic factor over a long-term period.

## 1. Introduction

Acute kidney injury (AKI) after cardiac surgery (AKI-CS) is a common complication, with a high incidence rate of up to 30% [1], and is a strong risk factor for increasing mortality [2]. Andrea et al. reported that even minimal changes in serum creatinine within 48 h after cardiac surgery increased mortality [3]. However, in cardiac surgery with cardiopulmonary bypass (CPB), serum creatinine immediately after surgery was normally decreased due to fluid balance. Therefore, a certain period of time must pass before serum creatinine is maximally elevated after the initiation of the diuretic phase. Failure to recognize AKI because of fluid overload may lead to a relative increase in mortality in those with fluid overload who (using conventional creatinine criteria) do not have AKI or have mild AKI, since minimal increases in creatinine are associated with significant increases in mortality [3,4]. Renal injury itself further increases the risk of developing severe non-renal complications that may lead to death [5,6]. Our hypothesis is that minimal changes in serum creatinine after cardiac surgery with CPB might affect postoperative renal function. In our previous study, we calculated the CRR using pre- and postoperative creatinine values [7].

While CRR exhibits potential as a discerning predictor for the detection of acute kidney injury after cardiac surgery (AKI-CS), it is important to note that our previous study did not incorporate comprehensive patient background information or anesthetic details into the analytical framework. Moreover, our previous study failed to delineate the capacity of CRR in forecasting the long-term trajectory of renal function beyond the immediate postoperative period following AKI. Our evaluation of acute kidney injury (AKI) risk predicated upon perioperative creatinine (Cr) dynamics and subsequent therapeutic interventions, while pertinent in the immediate perioperative context, lacks enduring clinical significance unless protracted prognostic assessments are undertaken that encompass outcomes such as the need for continuous renal replacement therapy (CRRT), the need for hemodialysis (HD), or the development of chronic kidney disease (CKD). A comprehensive analysis extending beyond the perioperative period is imperative to provide meaningful clinical relevance to predictive studies concerning AKI risk.

Therefore, in this study, we re-evaluated the reliability of CRR for short- and long-term outcomes by incorporating more perioperative information. The primary objective of this study was to determine the predictors of perioperative AKI-CS after surgery using perioperative information. The secondary objective was to determine whether perioperative serum creatinine (sCr) transition and AKI-CS affect the long-term outcomes of renal function.

Our overall objective in this study was to identify the pathogenic factors contributing to AKI and to assess the long-term prognosis of CRR, a predictive model that facilitates early intervention and prompt treatment measures.

Our model has been meticulously designed with the explicit intention of adaptability, and its simplicity may reduce reliance on biomarkers.

The rationale behind our approach lies in the belief that rapid identification of AKI risk within a clinical setting provides clear benefits, facilitating timely intervention and pre-emptive vigilance measures.

## 2. Materials and Methods

### 2.1. Study Population

This study was approved by the institutional review board at Yokohama City University Hospital (B130905027) and was registered at the University Medical Information Network (UMIN ID: 000011927). As this study is retrospective, patient consent was not required. We retrospectively collected data from patients who underwent cardiac surgery, such as coronary artery bypass grafting and valve replacement, with cardiopulmonary bypass at Yokohama City University Hospital between 2009 and 2014. In this study, patients undergoing elective surgery were included and patients undergoing emergency surgery, such as surgery for acute aortic dissection, were excluded.

We collected medical information from 299 patients, 74 of whom were excluded due to insufficient data. Therefore, 225 patients were included in the present study. A total of 124 patients were excluded from the evaluation due to insufficient data and preoperative CKD, and 101 patients were enrolled for the analysis of CKD (Figure 1).

### 2.2. Factor Measurement

In the preoperative patient data, information regarding age, hypertension, diabetes mellitus, renal function, heart disease, stroke, and respiratory function were collected. The anesthetic time, operation time, CPB time, aortic clamp time, transfusion, and perioperative creatinine values were also collected as operative information. Serum creatinine was measured immediately after surgery in the operation room or intensive care unit (ICU) and was compared with preoperative levels.

AKI was defined by the AKIN and RIFLE criteria using the elevation of sCr from baseline within 48 h after surgery. This was determined using the following two methods: (1) increment of creatinine ≥ 0.3 mg/dl and (2) increment of creatinine ≥ 150% [8]. CKD was defined as an estimated glomerular filtration rate (eGFR) < 60 mL/min per 1.73 m^2^ [9]. The eGFR, as defined by the Japanese Society of Nephrology and Pharmacotherapy, was calculated based on CRE, age, and sex [9]:eGFR(male): 194 × Cre^−1.094^ × Age^−0.287^
eGFR(female): eGFR(male) × 0.739

In our previous study, CRR was identified as a predictor of AKI-CS in cardiac surgery with CPB. CRR was defined as the perioperative change ratio of serum creatinine pre- and post-CPB. During CPB, blood dilution occurred due to the CPB priming volume. Therefore, serum creatinine levels fluctuate during the perioperative state. The perioperative CRR was calculated using pre- and postoperative creatinine values as follows:CRR = {(Post sCr − Pre sCr)/Pre sCr} × 100
where Post sCr represents the postoperative creatinine value and Pre sCr represents the preoperative creatinine value. In the employed computational algorithm, the retention of the negative sign is deliberate. Notably, when the preoperative creatinine level (PreCr) is elevated and the postoperative creatinine level (Post Cr) is diminished, the percentage change in creatinine is expressed as a percentage. This negative notation is indicative of a percentage reduction, aligning with the established presumption in this investigation that an affirmative elevation in postoperative creatinine serves as a marker for acute kidney injury (AKI) risk. Consequently, a CRR characterized by a negative sign was interpreted as predictive of a diminished AKI risk in this study.

The patients were divided into two groups: an AKI-CS group and a non-AKI-CS group, according to the AKIN and RIFLE criteria. Patients with insufficient data were excluded from this study. The predictors of AKI after cardiac surgery were determined using perioperative information. In previous studies on AKI-CS, some risk factors have been identified (e.g., age, female gender, cardiac failure, a history of cardiac surgery, chronic obstructive pulmonary disease, peripheral arterial disease, diabetes mellitus, renal function injury, emergency, and the use of intra-aortic balloon pumping) [2,3,4,10,11,12,13,14]. We used the receiver operating characteristic curve to determine the thresholds as predictors of CRR. The analysis was performed using the AKI-CS risk factors and CRR. APACHE II was calculated from 12 admission physiological variables, including the acute physiology score, the patient’s age, and their chronic health status.

Furthermore, we analyzed long-term kidney function. Chronic renal function was also evaluated using eGFR calculated using the modification of diet in renal disease (MDRD) at 6 months after operation. The CRR and perioperative AKI-CS incidence were evaluated as reliable predictors of chronic kidney disease.

### 2.3. Statistics

The quantitative analysis of background and perioperative information was described as the median within the interquartile range. A univariate analysis of patient information was performed between the AKI-CS and non-AKI-CS groups using the following method: Categorical data were analyzed using Fisher’s exact test, and continuous data were analyzed using the Mann–Whitney U test. The area under the ROC curve was calculated to assess the accuracy of a CRR < 20% in the diagnosing AKI. Various risk factors influence the incidence of AKI-CS [10]. A multivariate logistical analysis was performed to determine the predictors of AKI incidence. Potential predictor variables were selected by creating Directed Acyclic Graphs based on the risk factors reported in previous studies and using clinical experience (Figure 2).

An Akaike’s information criterion (AIC) stepwise procedure was used to select the final confounding variables for multivariate logistical analysis with the lowest value of the AIC.

For the two-tailed tests, a *p* value < 0.05 was considered statistically significant. Statistical analyses were performed using GraphPad Prism 6 for Mac OS X version 6.9b (GraphPad Software, Inc., San Diego, CA, USA) and R 2.13.0 statistical software (R Foundation for Statistical Computing, Vienna, Austria).

## 3. Results

We collected medical information from 299 patients, 74 of whom were excluded due to insufficient data (e.g., preoperative creatinine values and operation data). Therefore, 225 patients were included in the present study. The patients were divided into two groups: 97 patients (43%) in the AKI-CS group and 128 patients (57%) in the non-AKI-CS group. There were 33 patients (34%) who had preoperative CKD in the AKI group (24 male and 9 female) and 57 patients (44%) who had preoperative CKD in the non-AKI group (36 male and 21 female). Three patients each in the AKI and non-AKI groups required CRRT or HD after surgery.

The area under the receiver operating characteristic curve for AKI-CS prediction was 0.714 (0.650–0.778, *p* < 0.001). A CRR of 15% and 20% served as optimal threshold values with good diagnostic power (Figure 3) [7].

The area under the receiver operating characteristic curve for AKI prediction was 0.714 (*p* < 0.001).

CRR < 20% exhibited a sensitivity of 93.6%, a specificity of 28.3%, and a likelihood ratio of 1.31.

CRR < 15% exhibited a sensitivity of 88.6%, a specificity of 34.0%, and a likelihood ratio of 1.34.

ROC: receiver operating characteristic, CRR: creatinine reduction ratio, AKI: acute kidney injury.

### Univariate Analysis

Statistically significant disparities were observed between the AKI and non-AKI cohorts in several parameters, including surgical duration, cardiopulmonary bypass (CPB) duration, aortic cross-clamp duration, platelet transfusion, and distinctions in CRR values below 20% and 15% (Table 1).

## 4. Hospital Outcome and Long-Term Results

No statistically significant differences were observed in the APACHE II scores, ICU duration, and hospital stay between the two groups. Nevertheless, a noteworthy distinction emerged in the incidence of de novo chronic kidney disease (CKD) development among patients without a pre-existing history of renal dysfunction, suggesting a significant disparity between the two cohorts (Table 2).

In total, 124 patients were excluded from the evaluation due to insufficient data and preoperative CKD. A total of 101 patients with preoperative normal renal function were assessed for newly developed CKD in the postoperative period of 6 months. In the patients with normal renal function before surgery, the rate of development of CKD was higher in the AKI-CS group compared to the non-AKI-CS group (35.3% [18/51] vs. 10% [5/50], *p* = 0.004).

### Multivariate Analysis

In the multivariate logistical analysis, CRR < 15% was associated with the incidence of perioperative AKI-CS compared to CRR < 20% (adjusted odds ratio: 0.90 [0.87–0.93], *p* < 0.001) (Table 3).

The diagnostic power of CRR at thresholds of 20% and 15% was assessed for the incidence of AKI, new onset chronic kidney disease (CKD) with a CRR < 20% and a CRR < 15%, and perioperative AKI. In the comprehensive evaluation of the 225 patients included in this study, CRR, specifically at thresholds of <20% and <15%, emerged as a promising prognostic factor, demonstrating a sensitivity of 87.6% and 82.5%, a specificity of 33.6% and 46.9%, positive predictive values of 0.50 and 0.54, and negative predictive values of 0.78 and 0.78, respectively. Furthermore, the relative risk of AKI incidence associated with a CRR < 20% was 4.58 (95% CI: 1.79–11.7, *p* value < 0.001), and the relative risk of AKI incidence associated with a CRR < 15% was 2.45 (95% CI: 1.57–3.82, *p* value < 0.001) (Table 4).

## 5. Discussion

It has been established that renal function injury affects prognosis after cardiac surgery [2]. Several studies have reported the risk factors for AKI, for example, female gender, reduced left ventricular function, the presence of congestive heart failure, diabetes, chronic obstructive pulmonary disease, and elevated preoperative serum creatinine [15]. Even a slight increase in serum creatinine (<0.3 m/dl) may increase the mortality rate [3]. Blood flow reduction may lead to cellular injury due to an imbalance between oxygen delivery and demand [16], but it is not clear why a decrease in sCr protects against AKI. Our previous study reported that CRR may be associated with AKI-CS. In this study, we reconsidered the perioperative information, including CRR, and evaluated CRR as a predictor of long-term renal injury. In our data, the multivariate logistical analysis showed that a lower CRR is associated with perioperative AKI.

Despite the odds ratio for acute kidney injury (AKI) being 0.9, CRR emerges as a valuable predictive metric for AKI, a condition that may culminate in CKD. Consequently, CRR serves as a discerning prognostic indicator, facilitating the prospect of early intervention in the initial stages of AKI, thereby contributing to informed considerations of therapeutic strategies. To avoid the development of AKI, low-dose ANP and fenoldopam are recommended based on low-quality evidence. Hydroxyethyl starches and hyperchloremic solutions affect kidney function, so we should choose other solutions [17].

It is unclear whether changes in perioperative creatinine may lead to CKD. Brown et al. studied the relationship between the duration of AKI and prognosis after cardiac surgery. They reported that the mortality rate over a postoperative period of 5 years is 1.66 times higher for patients experiencing AKI for as little as 1–2 days [17]. AKI patients who recover to normal renal function before discharge have higher mortality rates than non-AKI patients [18]. In our study, there were no significant differences in CRR in terms of the rate of new onset CKD since the patients without data for a postoperative period of 6 months were excluded. CKD is defined by a decline in renal function over a period of at least 3 months, so we evaluated follow-up only up to 6 months in this study due to the timing of outpatient visits. However, renal function decline and dialysis were not evaluated after 6 months postoperatively, which we consider a limitation of this study. However, in the patients with normal renal function before surgery, the rate of development of CKD was higher in the AKI-CS group compared to the non-AKI-CS group. More than 90% of patients without AKI did not progress to CKD over a long period after surgery, and AKI is one of the risk factors for CKD development. CRR is effective for predicting AKI in the early period, so it is important to consider treatment in that period using CRR before new laboratory data become available.

## 6. Limitations

There are several limitations to this research. Urine output was not included in this study. Kama et al. reported that using the RIFLE criteria without considering the urine output criteria significantly underestimated the risk of AKI and delayed the diagnosis of AKI in critically ill patients [19]. In the period after cardiac surgery, urine output was strongly affected by the administration of diuretics during anesthesia and cardiopulmonary bypass. Since the diagnosis of AKI using decreased urine output requires adequate fluid volume and the absence of urinary tract obstruction, we excluded urine output from our definition of AKI. However, accurate monitoring of urine output may allow for an earlier diagnosis of acute kidney injury.

We calculated renal function using MDRD, which is affected by the patient’s age and serum biochemistry (Cre, BUN, and Alb). Immediately after surgery, eGFR is not used to evaluate acute kidney injury because of high fluid volumes due to transfusion. In contrast, eGFR is used as a diagnostic criterion for CKD because 6 months have passed since the operation. The biochemistry is variable after cardiac surgery. Moreover, the presence of diverse confounding elements, particularly in relation to Cr fluctuations and the incidence of AKI, introduces complexity in interpreting the CRR. The challenge of entirely mitigating collider bias emerges as a noteworthy limitation inherent in the present study. Creatinine is affected by fluid volume, so the effect of intraoperative fluid volume on the evaluation of renal function is significant. Furthermore, the evaluation of creatinine is more difficult in cardiac surgery because the in–out balance tends to change more widely than in other surgeries. We think that the infusion volume is easily influenced by the duration of the surgery, and further evaluation of each surgical procedure with an increased number of cases is necessary in the future. Although this study was conducted only in cardiac surgery, we would like to evaluate the effectiveness of CRR in other surgical procedures and in sepsis in further studies.

The present study was conducted as a single-center retrospective cohort study, and we collected information about the different types of surgery (only valve, CABG, congenital cardiac disease, and aortic surgery) and numerous patients who were excluded due to a lack of complete data. Furthermore, Chew et al. reported that the risk of AKI incidence differed according to race [20], and this may explain why the rate of AKI in our study was higher than in other studies. Furthermore, these results may not be adapted to another medical center. In future multicenter studies, we would like to limit the type of surgery and accurately set the time of postoperative blood sampling for evaluation. Also, we would like to investigate the short- and long-term changes that occur in patients at risk of acute kidney injury concerning infusion loading and the avoidance of drugs associated with a risk of kidney injury. Since this was a retrospective study, there were many cases in which the patient background items were not entered and the timing of postoperative blood sampling was not defined. In particular, we found that the postoperative blood sampling time was affected by the end time of the surgery in some cases, so we think that the timing of postoperative blood sampling needs to be defined at the time of data collection.

Lacquaniti et al. evaluated the role of proadrenomedullin (a urine tissue inhibitor of metalloproteinase 2 insulin-like growth-factor-binding protein 7 (TIMP2 IGFBP7)) and mid-region proadrenomedullin (MR-proADM) in predicting AKI in patients with sepsis after cardiac surgery [21]. They reported that these biomarkers are useful for predicting AKI in patients with sepsis after cardiac surgery.

Our study was retrospective, and the measurement of biomarkers was not performed according to the current clinical protocol. Therefore, biomarkers related to kidney function, such as urine tissue inhibitor of metalloproteinase 2 insulin-like growth-factor-binding protein 7 (TIMP2 IGFBP7) and neutrophil gelatinase-associated lipocalin (NGAL), were not included in this study. Gombert et al. reported that TIMP-2 and IGFBP7 are useful in predicting AKI, which leads to temporary renal replacement therapy after thoracoabdominal surgery [22]. However, the target patients in their study underwent open repair and endovascular repair; therefore, these results cannot be applied to AKI prediction after CPB. Lakhal et al. compared the usefulness of measuring small changes in creatinine (Cr) with biomarkers, such as TIMP2, IGFBP7, and NGAL, in predicting AKI after cardiac surgery. Their results indicated that the accuracy of predicting AKI through small changes in Cr was better than that achieved with these biomarkers [23]. NGAL measurements in future studies could be compared with diagnoses and predictions using creatinine. On the other hand, NGAL is affected by urinary infections and other factors, and more research is needed to determine whether NGAL can be applied in the perioperative period.

Therefore, the limitation inherent in our study is the absence of kidney function biomarkers. Additionally, given the specific focus on cardiac surgery patients undergoing cardiopulmonary bypass, the generalizability of our findings to diverse pathological contexts remains undetermined. While our investigation demonstrates potential applicability in cardiac surgery scenarios necessitating substantial fluid transfusions, such as sepsis and traumatic diseases, further research is imperative to ascertain the efficacy of CRR in these distinct medical conditions. However, this concept could be implemented at any other center due to its simple method of predicting AKI.

There are two advantages of using CRR. First, we can predict AKI by conducting blood tests immediately after surgery. In the conventional method, a long time is needed to define AKI because it is characterized by an increase in sCr within 24 to 48 h after the operation. However, we can predict AKI using CRR at the conclusion of the operation, which enables the consideration of treatment in the early period. Secondly, CRR can only be calculated using perioperative creatinine values, which can be obtained very easily and at low cost. Most patients are checked by conducting blood tests after surgery, so additional testing is not required.

The early prediction of AKI allows for earlier therapeutic intervention. However, there are few effective treatments for AKI, for example, to avoid HES or hyperchloremic solutions in elderly patients. Avoiding the use of nephrotoxic drugs can reduce the risk of renal function failure. However, there are cases in postoperative cardiac care where the use of nephrotoxic drugs is unavoidable, and we consider this to be a limitation. In the future, we would like to conduct a prospective study of the relationship between CRR and the risks of perioperative AKI.

## 7. Conclusions

CRR may be a useful predictor of perioperative AKI after cardiac surgery involving cardiopulmonary bypass, but it may not be useful for assessing the long-term outcomes of renal function. Further studies are required to verify our findings.

## Figures and Tables

**Figure 1 jcm-13-00009-f001:**
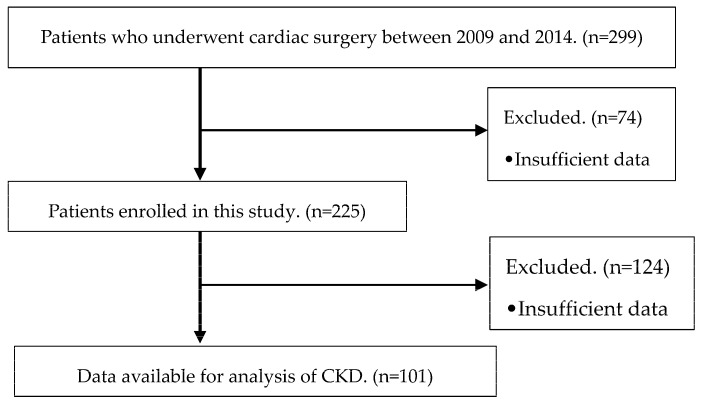
Inclusion criteria and exclusion criteria for patients in this study after cardiac surgery.

**Figure 2 jcm-13-00009-f002:**
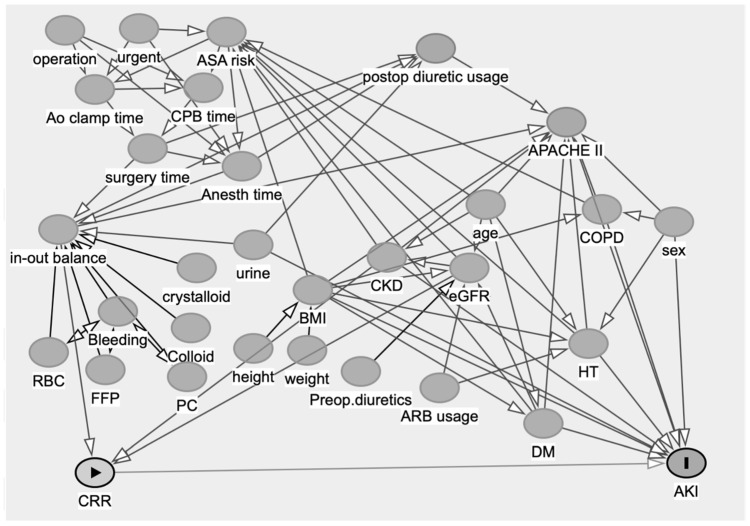
Acyclic Graphs based on the risk factors reported in previous studies and clinical experience.

**Figure 3 jcm-13-00009-f003:**
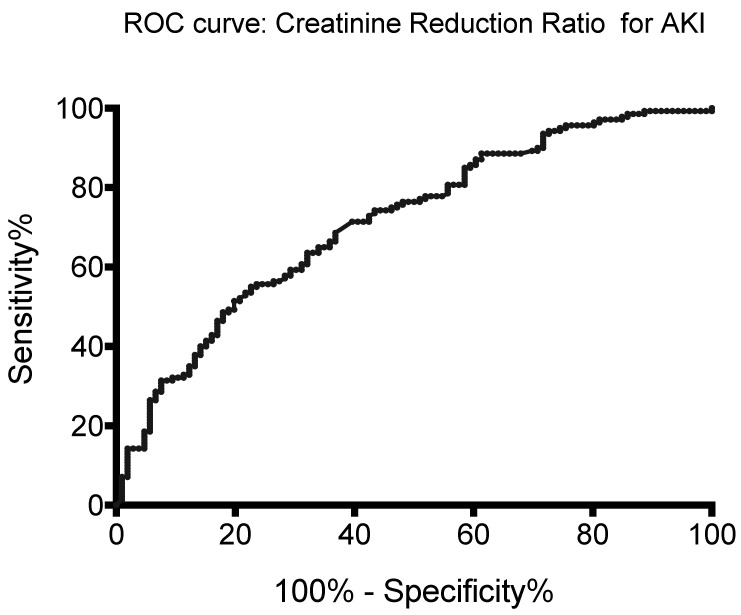
Area under the ROC curve of the creatinine reduction ratio for AKI prediction.

**Table 1 jcm-13-00009-t001:** Background and perioperative information in the total group, the AKI group, and the non-AKI group.

	Total (*n* = 225)	AKI (*n* = 97)	Non-AKI (*n* = 128)	*p* Value
Age (yr)	66 (49–75)	67 (50–74)	64.5 (48–75.25)	0.69
Sex (male, %)	148 (65.8%)	65 (67.0%)	83 (64.8%)	0.78
Height (m)	1.64 (1.56–1.7)	1.63 (1.55–1.68)	17 (11.6%)	0.57
Weight (kg)	59.2 (49.5–66.5)	58.8 (49.23–66.20)	56 (38.3%)	0.90
Body mass index (BMI)	22.1 (19.4–24.3)	22.32 (19.59–24.60)	30 (20.5%)	0.54
ASA risk	3 (2–3)	3 (2–3)	3(2–3)	0.86
Urgent (*n*, %)	29 (12.8%)	8 (7.5%)	20 (13.7%)	0.16
HT (*n*, %)	89 (39.5%)	37 (38.1%)	52 (40.6%)	0.78
DM (*n*, %)	34(15.1%)	18(18.5%)	16 (12.5%)	0.26
COPD (*n*, %)	1 (0.04%)	1 (1.03%)	0 (0%)	0.43
Preop. diuretics (*n*, %)	61 (27.1%)	28 (28.8%)	33 (25.7%)	0.65
CKD (*n*, %)	90 (40%)	33 (34%)	57 (44%)	0.05
eGFR	66.7 (50.1–85.4)	68.6 (54.2–85.8)	63.5 (46.6–84.4)	0.11
ARB usage (*n*, %)	70 (31.1%)	33 (34.0%)	37 (28.9%)	0.51
Operation				
CABG (*n*, %)	32 (27.1%)	15 (14.2%)	17 (11.6%)	0.57
Valve (*n*, %)	95 (42.2%)	39 (36.8%)	56 (38.3%)	0.90
Great artery (*n*, %)	56 (24.8%)	26 (24.5%)	30 (20.5%)	0.54
Others (congenital) (*n*, %)	41 (18.2%)	18 (16.5%)	23 (15.8%)	0.86
Combined (*n*, %)	28 (12.4%)	8 (7.5%)	20 (13.7%)	0.16
Surgery time (hour)	7.63 (5.96–9.83)	8.21 (6.20–10.25)	7.38 (5.63–9.41)	0.003
CPB time (hour)	3.61 (2.71–4.68)	3.9 (3.05–4.91)	3.31 (2.53–4.4)	0.03
Ao clamp (hour)	2.38 (1.55–3.13)	2.61 (1.81–3.43)	2.25 (1.45–2.86)	0.03
Anesth time (hour)	9.46 (7.73–11.75)	10.2 (8.21–12.1)	9.15 (7.46–11.3)	0.004
In–out balance (L)	2.70 (0.899–5.18)	2.48 (0.64–4.70)	2.95 (1.18–5.54)	0.51
RBC (mL)	1120 (560–1960)	1120 (560–1680)	1120 (560–1960)	0.37
PC (mL)	500 (250–750)	500 (250–750)	250 (500–500)	0.02
FFP (mL)	1200 (720–2160)	1440 (720–2160)	960 (720–2010)	0.58
Crystalloid (L)	3.75 (2.4–12.38)	3.3 (2.3–7.33)	4.5 (2.4–13.2)	0.08
Colloid (ml)	550 (500–1000)	500 (500–1000)	650 (500–1000)	0.92
Bleeding (L)	1.89 (1.06–3.90)	1.84 (0.791–3.97)	1.91 (1.1–3.65)	0.57
Urine (mL)	2560 (1660–3450)	2270 (1657–3450)	2655 (1665–3432)	0.31
Diuretic usage (*n*, %)	38 (16.8%)	21 (34.9%)	17 (13.3%)	0.08
CRR < 20% (*n*, %)	170 (75.6%)	85 (87.6%)	85 (66.4%)	<0.001
CRR < 15% (*n*, %)	148 (83.5%)	80 (82.4%)	68 (53.1%)	<0.001

ASA: American Society of Anesthesiologists, HT: hypertension, DM: diabetes Mellitus, COPD: chronic obstructive pulmonary disease, CKD: chronic kidney disease, ARB: angiotensin II receptor blocker, CABG: coronary artery bypass graft, CPB: cardiopulmonary bypass, Ao: aorta, RBC: red blood cell, PC: platelet concentrate, FFP: fresh frozen plasma, CRR: creatinine reduction ratio.

**Table 2 jcm-13-00009-t002:** Postoperative information between AKI and non-AKI groups after cardiac surgery with cardiopulmonary bypass.

	AKI-CS Group (*n* = 97)	Non-AKI-CS Group (*n* = 128)	*p* Value
APACHE II	17.6 ± 8.0	18.1 ± 7.7	0.58
Ventilator Free Days (day)	4.8 ± 4.2	4.2 ± 2.9	0.17
ICU duration (day)	4.8 ± 4.2	4.2 ± 2.9	0.17
Hospital Stay (day)	42.0 ± 90.5	31.2 ± 22.1	0.17
	AKI-CS group 6 month (*n* = 51)	Non-AKI-CS group 6 month (*n* = 50)	
CKD development (*n*, %)	18 (18/51, 35.3%)	5 (5/50, 10%)	0.004

APACHE II: acute physiology and chronic health evaluation II, CKD: chronic kidney disease. CKD development: 101 patients with preoperative normal renal function were assessed for newly developed chronic kidney disease in the postoperative period of 6 months.

**Table 3 jcm-13-00009-t003:** Multivariate logistical regression analysis of prognostic factors for perioperative AKI incidence after cardiac surgery.

	Adjusted Odds Ratio	95% Confidence Interval	*p* Value
APACH II	0.95	0.90–1.00	0.07
CPB time	2.32	1.45–3.73	<0.001
CRR	0.90	0.87–0.93	<0.001
Preop. eGFR	1.02	1.00–1.03	0.02
Surgery time	1.29	1.04–1.61	0.02

APACHE II: acute physiology and chronic health evaluation II, CPB: cardiopulmonary bypass, CRR: creatinine reduction ratio, eGFR: estimated glomerular filtration ratio.

**Table 4 jcm-13-00009-t004:** Diagnostic power of a creatinine reduction ratio of 20% and 15% for AKI incidence and diagnostic power of a CRR < 20%, CRR < 15%, and perioperative AKI for new CKD development.

Predictor	CRR < 20%	CRR < 15%	CRR < 20%	CRR < 15%	Perioperative AKI
Outcome	AKI incidence	AKI incidence	New CKD development	New CKD development	New CKD development
	AKI vs. non-AKI (*n* = 225) 97 vs. 128	AKI vs. non-AKI (*n* = 225) 97 vs. 128	CKD vs. non-CKD (*n* = 101) 23 vs. 78	CKD vs. non-CKD (*n* = 101) 23 vs. 78	CKD vs. non-CKD (*n* = 101) 23 vs. 78
Unadjusted odds ratio	8.08 (2.75–23.7)	4.15 (2.22–7.78)	0.36 (0.07–1.74)	0.86 (0.25–2.99)	4.91 (1.65–14.57)
Sensitivity	87.6% (79.4–93.4)	82.5% (73.4–89.5)	86.7% (66.4–97.2)	82.6% (61.2–95.1)	78.3% (56.3–92.5)
Specificity	33.6% (25.5–42.5)	46.9% (38.0–55.9)	5.1% (1.4–12.6)	15.4% (8.2–25.3)	57.7% (46.0–68.8)
PPV	0.50 (0.42–0.57)	0.54 (0.46–0.62)	0.21 (0.13–0.31)	0.22 (0.14–0.33)	0.35 (0.22–0.50)
NPV	0.78 (0.65–0.88)	0.78 (0.67–0.87)	0.57 (0.18–0.90)	0.75 (0.48–0.93)	0.90 (0.78–0.97)
Relative risk	4.58 (1.79–11.7)	2.45 (1.57–3.82)	0.49 (0.19–1.27)	0.89 (0.35–2.28)	3.53 (1.42–8.78)
*p* value	<0.001	<0.001	0.19	0.76	0.004

CRR: creatinine reduction ratio, CKD: chronic kidney disease, AKI: acute kidney injury, PPV: positive predictive value, NPV: negative predictive value.

## Data Availability

The data are available from the corresponding author on reasonable request.

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
