# Peer review of "Creatinine Reduction Ratio Is a Prognostic Factor for Acute Kidney Injury following Cardiac Surgery with Cardiopulmonary Bypass: A Single-Center Retrospective Cohort Study"

_jcm, 2023, doi:10.3390/jcm13010009_

Round 1

Reviewer 1 Report

Comments and Suggestions for Authors

Dear Authors,

It is a very great pleasure for me to review this paper

The topic are very interesting.

I have some questions for you:

1. Are there  patients who have undergone acute aortic dissection in this study?

2. Why has the role the biomarkers as TIMP2*IGFBP7 not been evaluated?

3 How many patients with AKI -CS underwent CRRT?

I suggest you to compare these biomarkers with CCR to predict AKI-CS and especially to predict who will undergo CRRT. 

In the discussion. you shoul cite and briefly discuss the following reference:

 - Lacquaniti A et al Journal of Clinical Medicine Open Access Sepsis after Cardiac Surgery: The Roles of Tissue Inhibitor Metalloproteinase-2, Insulin-like Growth Factor Binding Protein-7, and Mid-Regional Pro-Adrenomedullin. 2023; 12:16

Author Response

Thank you for your comment.
We revised the manuscript based on the comments received.

Below are the changes I made based on your comments.

Comment 1. Are there patients who have undergone acute aortic dissection in this study?

We revised according to the comments and included the following sentence in the paper.

In this study, patients undergoing elective surgery were included, excluding emergency surgeries such as acute aortic dissection.

Comment 2. Why has the role the biomarkers as TIMP2*IGFBP7 not been evaluated?

We revised according to the comments and included the following sentence in the paper.

Lacquaniti A et al. evaluated the role of proadrenomedullin (urine tissue inhibitor of metalloproteinase 2 insulin-like growth factor-binding protein 7 (TIMP2 IGFBP7)) and midregion proadrenomedullin (MR-proADM) in predicting AKI in patients with sepsis after cardiac surgery.

They reported that these biomarkers are useful for predicting AKI in patients with sepsis after cardiac surgery.

Our study was a retrospective study, and the measurement of biomarkers was not performed according to the clinical protocol to date. Therefore, biomarkers related to kidney function such as TIMP2 IGFBP7, and neutrophil gelatinase-associated lipocalin(NGAL)were not included in this study.

Gombert et al. have found that TIMP-2 and IGFBP7 are useful in predicting AKI, which leads to temporary renal replacement therapy after thoracoabdominal surgery, but the target patients include open repair and endovascular repair, and they are unable to perform post-cardiopulmonary bypass surgery. cannot necessarily be applied to AKI predictions.

Lakhal et al. compared the usefulness of measuring small changes in Cr with biomarkers such as TIMP2, IGFBP7, and NGAL in predicting AKI after cardiac surgery. The accuracy of  predicting AKI by Cr small changes was better than these biomarkers.

Therefore, our research limitation is the absence of biomarkers for renal function. However, this concept could be applied at any other center due to its simple method of predicting AKI.

Comment 3.  How many patients with AKI -CS underwent CRRT?

We revised according to the comments and included the following sentence in the paper.

Three patients each in the AKI and non-AKI groups were required continuous renal replacement therapy (CRRT) or Hemodialysis (HD) after surgery.

Comment 4. I suggest you to compare these biomarkers with CCR to predict AKI-CS and especially to predict who will undergo CRRT.

In the discussion. you should cite and briefly discuss the following reference:

 - Lacquaniti A et al Journal of Clinical Medicine Open Access Sepsis after Cardiac Surgery: The Roles of Tissue Inhibitor Metalloproteinase-2, Insulin-like Growth Factor Binding Protein-7, and Mid-Regional Pro-Adrenomedullin. 2023; 12:16

I have also included this reference in conjunction with the above question.

Could you check answer to question 2.

Reviewer 2 Report

Comments and Suggestions for Authors

I found the study conducted by my colleagues to be exceptionally interesting, well-executed, and practical. The calculation of the CRR was straightforward and easy to interpret, and it demonstrated the value of the CRR as a predictor of AKI in patients who underwent cardiopulmonary bypass. Although the study was retrospective and conducted through electronic file search, it was well-designed and detailed. 

I have a few recommendations for some minor revisions, which I believe could add value to the study. Firstly, I would like the authors to clarify the CRR formula. In some cases, the formula produced a negative sign (e.g. Cr pre 0.8, Cr post 1 mg). I would like to know if the negative sign is implicitly removed and whether we should treat the percentage as an absolute number. 

Secondly, I suggest that the authors discuss the issue of acute kidney injury and CRR in more detail. Additionally, I recommend that they specify whether there are any studies on this score for other conditions of kidney damage during surgical or medical treatments that use high volumes of infusions. If such studies are not yet available, they could hypothesize the possible usefulness of the CRR in other fields and what they might be. 

Finally, I recommend that the authors create a paragraph that details the limitations of the study. This should include the retrospective nature of the study and the issue of numerous patients who were excluded due to lack of complete data.

Comments on the Quality of English Language

Minor editing of English language required

Author Response

Thank you for your comment.
We revised the manuscript based on the comments received.

Below are the changes I made based on your comments.

Comment 1.

I have a few recommendations for some minor revisions, which I believe could add value to the study. Firstly, I would like the authors to clarify the CRR formula. In some cases, the formula produced a negative sign (e.g. Cr pre 0.8, Cr post 1 mg). I would like to know if the negative sign is implicitly removed and whether we should treat the percentage as an absolute number. 

We added following sentence in the method section according to your comment.

 In the employed computational algorithm, the retention of the negative sign is deliberate. Notably, when the preoperative creatinine level (PreCr) is elevated and the postoperative creatinine level (Post Cr) is diminished, the resulting calculation yields a Cardiovascular Risk Reduction (CRR) expressed as -〇〇%. This negative notation is indicative of a percentage reduction, aligning with the established presumption in this investigation that an affirmative elevation in postoperative creatinine serves as a marker for Acute Kidney Injury (AKI) risk. Consequently, a CRR characterized by a negative sign is construed in this study as predictive of a diminished AKI risk.

Comment 2.

Secondly, I suggest that the authors discuss the issue of acute kidney injury and CRR in more detail. Additionally, I recommend that they specify whether there are any studies on this score for other conditions of kidney damage during surgical or medical treatments that use high volumes of infusions. If such studies are not yet available, they could hypothesize the possible usefulness of the CRR in other fields and what they might be. 

Therefore, the limitation inherent in our study lies in the absence of kidney function biomarkers. Additionally, given the specific focus on cardiac surgery patients undergoing cardiopulmonary bypass, the generalizability of our findings to diverse pathological contexts remains undetermined. While our investigation demonstrates potential applicability in cardiac surgery scenarios necessitating substantial fluid transfusions, such as sepsis and traumatic diseases, further research endeavors are imperative to ascertain the efficacy of CRR in these distinct medical conditions. However, this concept could be applied at any other center due to its simple method of predicting AKI. 

Comment 3.

Finally, I recommend that the authors create a paragraph that details the limitations of the study. This should include the retrospective nature of the study and the issue of numerous patients who were excluded due to lack of complete data.

We created limitation section according to your comment, and added the issue due to lack of data.

This present study was conducted single center retrospective cohort study, and we collected different type of surgery (only valve, CABG, congenital cardiac disease, and aortic surgery) and numerous patients who were excluded due to lack of complete data.

Reviewer 3 Report

Comments and Suggestions for Authors

Dear Authors,

I have received an article that attempts to elucidate the relationship of creatinine reduction ratio (CRR) as a prognostic factor for AKI in cardiac surgery patients with cardiopulmonary bypass. However, there are several issues in this manuscript.

1) The abstract is not informative on its own. It is too lengthy on the background with minimal information in the results section. Furthermore, the primary aim of the study was the determination of AKI predictor, while in the main text it was the CRR.

2) Lines 56-58 seem to be a duplicate with the methodology section

3) The authors mentioned the previous study (citation number 4) and this study is an extension of that. However, I noticed that the previous study has 500+ participants. Please explain how that number is now only 225.

4) Lines 70-71 --> With so many independent variables, there is bound to be collider bias. Please evaluate these relationships with directed acylic graph (DAG)

5) The definition of AKI used in this study only constitutes the "Risk" part of RIFLE. Why was this used? The authors may argue that it is more sensitive towards AKI but it is not specific as it is only pre-AKI and hence defeats the purpose of this study.

6) CKD is defined by eGFR <60 for three months. Did this study ensure that the three months elapse before diagnosing CKD?

7) Please provide citations on how the formula of eGFR was quoted from

8) Please provide the CONSORT diagram of the patient selection

9) Why was there suddenly an ROC curve when the methodology did not mention so?

10) The likelihood ratio is too small, negligible even, for clinical utility. Please put this in the discussion.

11) Why was the urine only 2.56 ml in Table 1 for total patients?

12) Please explain more in greater details about APACHE II in the methodology section and how the values were derived.

13) The authors mention new CKD development. How was this observed since this was not mentioned in the methodology section.

14) Please include a limitation section

Comments on the Quality of English Language

Extensive editing is required

Author Response

Thank you for your comment.
We revised the manuscript based on the comments received.

Here attached the changes I made based on your comments.

Round 2

Reviewer 1 Report

Comments and Suggestions for Authors

Dear authors

I am satisfied to your efforts 

Author Response

Thank you for taking the time to review.

Following the comments of other reviewers, I completed the attached paper.

Reviewer 3 Report

Comments and Suggestions for Authors

Dear Authors,

I applaud the changes made as I can see the efforts towards change. However, there are still numerous unaddressed points:

1) The abstract has not been changed at all. Please refer on journal articles on how to write good abstracts.

2) Long-term prognosis is just an outcome which may address a research gap, however it cannot be included as a strength of the study. 

3) Please include the DAG in the manuscript. Also, a DAG needs to have connections, and it is acyclic. Sex and APACHE II are not connected to anywhere, please revise this as this shows the basic concept of this research is not strong enough

4) If the authors introduced a concept of cost-effectiveness, then a cost effectiveness analysis (CEA) needs to be done.

5) Line 108 --> What is -O O%?

6) Again, chronic kidney disease by definition can only be diagnosed after 6 months. However, the authors mention 3-6 months. How can they ensure that at 3-<6 months, the patients were reliably diagnosed as patients with chronic kidney disease?

Comments on the Quality of English Language

The English language still needs extensive editing

Author Response

Thank you for your thorough review.
I've made corrections according to comments.

Revised sentences are highlighted in red. The English proofreading was also done again.

I would appreciate it if you could check it out.
